# Antiretroviral Therapy-Induced Dysregulation of Gene Expression and Lipid Metabolism in HIV+ Patients: Beneficial Role of Antioxidant Phytochemicals

**DOI:** 10.3390/ijms23105592

**Published:** 2022-05-17

**Authors:** Angélica Saraí Jiménez-Osorio, Sinaí Jaen-Vega, Eduardo Fernández-Martínez, María Araceli Ortíz-Rodríguez, María Fernanda Martínez-Salazar, Reyna Cristina Jiménez-Sánchez, Olga Rocío Flores-Chávez, Esther Ramírez-Moreno, José Arias-Rico, Felipe Arteaga-García, Diego Estrada-Luna

**Affiliations:** 1Área Académica de Enfermería, Instituto de Ciencias de la Salud, Universidad Autónoma del Estado Hidalgo, Circuito Ex Hacienda La Concepción S/N, Carretera Pachuca-Actopan, San Agustín Tlaxiaca 42160, Mexico; angelica_jimenez@uaeh.edu.mx (A.S.J.-O.); ja369176@uaeh.edu.mx (S.J.-V.); jimenezs@uaeh.edu.mx (R.C.J.-S.); ofloresc@uaeh.edu.mx (O.R.F.-C.); jose_arias@uaeh.edu.mx (J.A.-R.); 2Laboratorio de Química Medicinal y Farmacología, Centro de Investigación en Biología de la Reproducción, Área Académica de Medicina, Instituto de Ciencias de la Salud, Universidad Autónoma del Estado de Hidalgo, Calle Dr. Eliseo Ramírez Ulloa no. 400, Col. Doctores, Pachuca Hidalgo 42090, Mexico; tomedyfm@hotmail.com; 3Facultad de Nutrición, Universidad Autónoma del Estado de Morelos, Iztaccíhuatl 100 Col. Los Volcanes, Cuernavaca 62350, Mexico; araceli.ortiz@uaem.mx; 4Facultad de Ciencias del Deporte, Facultad de Farmacia Universidad Autónoma del Estado de Morelos, Av. Universidad No. 1001 Col. Chamilpa, Cuernavaca 62209, Mexico; fernandamartinez@uaem.mx; 5Área Académica de Nutrición, Instituto de Ciencias de la Salud, Universidad Autónoma del Estado Hidalgo, Circuito Ex Hacienda La Concepción S/N, Carretera Pachuca-Actopan, San Agustín Tlaxiaca 42160, Mexico; esther_ramirez@uaeh.edu.mx; 6Coordinación de Enseñanza e Investigación, Hospital del Niño DIF Hidalgo, Carretera México-Pachuca km 82, Pachuca de Soto 42080, Mexico; felipearteagagarcia@gmail.com

**Keywords:** antiretroviral therapy, phytochemicals, lipid metabolism, genes, HIV, oxidative stress

## Abstract

Human immunodeficiency virus (HIV) infection has continued to be the subject of study since its discovery nearly 40 years ago. Significant advances in research and intake of antiretroviral therapy (ART) have slowed the progression and appearance of the disease symptoms and the incidence of concomitant diseases, which are the leading cause of death in HIV+ persons. However, the prolongation of ART is closely related to chronic degenerative diseases and pathologies caused by oxidative stress (OS) and alterations in lipid metabolism (increased cholesterol levels), both of which are conditions of ART. Therefore, recent research focuses on using natural therapies to diminish the effects of ART and HIV infection: regulating lipid metabolism and reducing OS status. The present review summarizes current information on OS and cholesterol metabolism in HIV+ persons and how the consumption of certain phytochemicals can modulate these. For this purpose, MEDLINE and SCOPUS databases were consulted to identify publications investigating HIV disease and natural therapies and their associated effects.

## 1. Introduction

Since the first case was reported in the 1980s, human immunodeficiency virus (HIV) infection has spread worldwide and become one of the leading causes of death [1]. In 2020, it was estimated that approximately 37.6 million people were infected with HIV. A total of 95.5% of them were adults, and 4.5% were children under 14 years of age; on account of the low accessibility of antiretroviral therapy (ART), children comprise the most vulnerable group to develop a comorbidity or die. Treatment with ART is critical at the initial stages of infection because it reduces the virus replication rate and keeps the viral load at undetectable levels [2]. The main antiretroviral drugs currently available and the most effective have been described as protease inhibitors (PIs), integrase inhibitors (INIs), and reverse transcriptase inhibitors (RTIs), which are divided into nucleoside analogs (NARTI or NRTI), nucleotide analogs (NtARTI or NtRTI) and non-nucleoside analogs (NNRTI) [3]. However, the use of these antiretrovirals may increase oxidative stress (OS) because, together with the natural course of the disease, an increase in reactive oxygen species (ROS) and other prooxidant substances have been found in the organism [4,5,6]. OS may activate pro-inflammatory signaling pathways, considered risk factors for various chronic degenerative diseases [7,8], the main ones linked to cardiovascular events [9,10,11]. Likewise, OS significantly decreases the effectiveness of ART, enhancing the complication of the illness and the appearance of opportunistic conditions [12].

To reverse or delay the onset of those conditions in HIV patients, exacerbated by ART, supplementation with various phytochemicals could represent a therapeutic alternative to pharmacological treatments alone to reduce the cardiovascular risk and perhaps improve CD4 lymphocyte counts [13].

## 2. HIV

HIV is a retrovirus of the lentivirus genus that causes a slow and progressive reduction of the immune system due to viral replication, mainly in the CD4 lymphocyte cells [14].

After entering the body, the virus infects host cells by binding to the CD4 receptor, and chemokine CCR5 or CXCR4 co-receptors found mainly on T lymphocytes (Figure 1) and macrophages, dendritic cells, and monocytes. Inside the cell, single-stranded RNA is released, which will serve as a template for synthesizing double-stranded viral DNA by retrotranscription, allowing the virus to enter the nucleus helped by integrase to place its genetic material with that of the host cell. Once the new viral RNA is formed, it is used as genomic RNA to form viral proteins that will be mobilized back to the cell membrane, giving rise to an immature (non-infectious) virus; this leads to the release of proteases from the virus for the degradation of the long-chain polypeptides, generating mature viruses that will allow the virus to spread in the organism [15,16,17].

HIV infection is branded by a series of stages with peculiar clinical evidence, such as the acute retroviral infection phase, which may be described as asymptomatic, although it can be accompanied by pharyngitis, fever, myalgias, and others. Likewise, gastrointestinal, dermatological, and neurological symptoms may also occur [18,19]. The asymptomatic phase is marked by being asymptomatic or presenting an adenoid syndrome with the presence of firm, but not woody, mobile, non-painful lymph nodes without changes in the overlying skin, occupying two or more adjacent regions [20,21]. During the AIDS phase, immunosuppression is exacerbated with a considerable loss of CD4 lymphocytes and significant viral replication, neoplasms, and opportunistic infection [22,23,24,25,26,27,28]. 

The initial treatment of HIV is essential to avoid the progression of the infection. However, only 59% of people with HIV receive ART and follow up the illness in a care center; other people abandon the service after being diagnosed and return when the illness is significantly advanced and has deteriorated the immune system; which translates into a significant increase in morbidity and mortality rates [29,30].

### 2.1. Antiretroviral Therapy (ART)

ART has increased the life expectancy in HIV patients such that it is similar to that of the general population, making HIV a treatable chronic disease from a multidisciplinary approach [23,31]. 

One of the most effective combinations is highly active antiretroviral therapy (HAART), where three or more drugs are used to suppress the viral load to undetectable levels, leading to immunological recovery in a shorter time. It is considered the most effective strategy for treating HIV and also concerning its cost–benefit ratio, since it reduces hospitalizations due to complications, and the incidence of opportunistic infections, improving the quality of life [32,33].

#### 2.1.1. Proteases Inhibitors

This group of drugs inhibits protease enzyme activity (Table 1), which prevents the cleavage of Gag and Gag-Pol proteins, causing the formation of non-infectious immature virions and following interruption of viral shedding [34,35,36].

#### 2.1.2. Nucleoside/Nucleotide Reverse Transcriptase Inhibitors (NRTIs)

NRTIs (Table 2) inhibit reverse transcriptase enzyme by a competitive mechanism with physiological nucleosides/nucleotides; NRTIs are incorporated into the viral DNA chain, interrupting its elongation and, as a consequence, viral replication [50,51].

#### 2.1.3. Non-Nucleoside Reverse Transcriptase Inhibitors (NNRTIs)

NNRTIs (Table 3) inhibit viral DNA synthesis by binding to an allosteric site on reverse transcriptase. They can be administered in conjunction with some NRTIs as initial therapy for HIV infection for replacement or alternative to PIs [68]. 

### 2.2. Effects of Antiretroviral Therapy on Lipid and Cholesterol Metabolism

In addition to HIV infection, the administration of PI-based ART or combined ART and the inflammation that occurs in the disease leads to changes in lipid metabolism that are associated with a rise in the incidence of metabolic risk factors, including insulin resistance, dyslipidemia, glucose intolerance, metabolic syndrome (MetS), lactic acid elevation, and some types of lipodystrophy [82,83,84,85,86,87]. As the disease progresses, CD4 lymphocytes induce the inflammatory response by increasing the levels of OS and the production of cytokines, which results in a decrease in the concentration of high-density lipoprotein cholesterol (HDL-C), altering the reverse cholesterol transport (RCT) and accumulating cholesterol in macrophages [88]. On the other hand, the proliferation of pro-inflammatory lipids as oxidized low-density lipoproteins (ox-LDL) are associated with the expression of inflammatory markers of the immune system as interleukin 1β, tumor necrosis factor alpha (TNF-α), and interleukin six, and that lasts even if viral replication is inhibited by ART. This pro-inflammatory state correlates with the development of thrombosis. Besides this, the increase in viral proteins and cytokines stimulates endothelial lipase activity leading to mitochondrial dysfunction, the production of prooxidant chemical species, insulin resistance, a decrease in adiponectin, and an increment in remnant lipoproteins [89].

In the same way, it has been established that HDL-C, both large (HDL2b and HDL2a) and small particles (HDL3a, HDL3b, and HDL3c), have been associated with cardiovascular risk in HIV patients. Although this process is exacerbated by ART, it continues to be unclear whether regulation of its chemical composition can modify changes in lipid metabolism and the effects produced by ART, which include rising OS and activation of pro-inflammatory pathways [90,91]; however, the increase in plasma HDL-C concentration does not reverse cardiovascular risk.

### 2.3. Effects of HIV Infection and ART on Lipid and Cholesterol Genes

Lipodystrophies are characterized by differing degrees of body fat loss, including a tendency to metabolic disturbances such as insulin resistance, diabetes, hypertriglyceridemia, and hepatic steatosis; their origin can be genetic or acquired. The two most common acquired types are generalized and partial lipodystrophy. Common subtypes are HAART-associated lipodystrophy syndrome (HAALS) in HIV patients and drug-induced localized lipodystrophy. Indeed, HAALS seems to be multifactorial and may occur after two or four years of HAART in patients administered with PIs or NRTIs [92]. As PIs inhibit the zinc metalloprotease STE24 (ZMPSTE24), the enzyme responsible for prelamin A processing, an accumulation of toxic farnesylated prelamin A may cause the dysregulation of transcription factors involved in adipogenesis and HIV-associated cardiomyopathy provoked by inflammation—perhaps due to the modulation of NFκB signaling via the DNA damage transducer ataxia telangiectasia mutated (ATM) [92,93]. Moreover, NRTIs inhibit the mitochondrial DNA (mtDNA) polymerase-γ transcription, leading to mtDNA depletion, which causes diverse pathologies, including lipodystrophy and hepatosteatosis mediated by pro-inflammatory cytokines [94]. These data suggest that targeting inflammation may be a helpful treatment approach.

Not only do the drugs used in ART cause metabolic disturbances, but HIV infection alone has also been strongly associated with an increased risk of cardiovascular diseases. Untreated HIV-infected persons present lower levels of HDL-C than HIV-negative people; HDL-C levels are lower after the acquisition of the virus [95,96]. It has been demonstrated in apoE−/− mice that the accumulation of abnormal metabolites such as oxLDL upregulates purinergic 2X7 receptor (P2X7R), nucleotide-binding oligomerization domain-like receptor protein 3 (NLRP3) inflammasome, and interleukin (IL)-1β expression during atherosclerotic plaque formation through protein kinase R (PKR) phosphorylation [97,98]. In HIV infection, HIV-1 transactivator of transcription protein (Tat) and HIV-1 protease increase NLRP3 inflammasome [99] and contribute to CD4+ T loss through pyroptosis [100]. Additionally, hypercholesterolemia increases caspase-1 activity in the coronary arterial endothelium of Nlrp3(+/+) mice through superoxide production, leading to the downregulation of endothelial nitric oxide synthase activity pyroptosis [101].

HDL-C also participates in the RCT from peripheral tissues into circulation and the liver, where cholesterol can be metabolized or eliminated [102]. It has been demonstrated in vitro that HIV infection via negative regulatory factor (Nef) protein harms the monocyte-macrophage cholesterol efflux by increasing the degradation of the ATP-binding cassette transporter-A1 (ABCA1), a crucial transporter of lipids and cholesterol from cells to extracellular apolipoprotein-A1 [102], but by up-regulating its mRNA [103]. In vivo, untreated HIV persons had lower HDL-C levels while ABCA1 mRNA was elevated compared to ART-treated patients and HIV-negative people.

Interestingly, the expression of genes related to cholesterol uptake (low-density lipoprotein receptor, LDLR, and scavenger receptor class B member-3, SCARB3, also known as cluster of differentiation-36, CD36), synthesis (3-hydroxy-3-methylglutaryl-CoA reductase, HMGCR), and regulation (sterol regulatory element-binding transcription factor 2 or SREBP2 and liver X receptor-alpha, LXRα) were significantly decreased in treated or untreated HIV infected patients in contrast to the HIV-negative group [96]. That study indicates that HIV and ART impact monocyte-macrophage cholesterol metabolism, which is vital for forming foam cells and atherosclerosis. Furthermore, a study on HIV treatment-experienced individuals revealed a dysregulation between SRBP2 and HMGCR and LDLR pathways, which may precede the clinical manifestation of ART-induced lipid metabolism derangement [104]. 

On the other hand, interferons (IFN) are recognized antiviral cytokines which up-regulate many interferon-stimulated genes (ISG). One of these ISG is cholesterol-25-hydroxylase (CH25H), which converts cholesterol to 25-hydroxycholesterol (25HC), a soluble antiviral factor since, in cultured cells, 25HC inhibits the growth of enveloped HIV by blocking membrane fusion with cells; besides, in humanized mice, the 25HC suppressed the HIV replication and T cell depletion [105]. Furthermore, there is strong evidence that sterol metabolism may confer natural resistance to HIV-1 infection in exposed seronegative people (HESN). For instance, peripheral blood mononuclear cells (PBMCs) and monocyte-derived macrophages (MDMs) were isolated from HESN and compared to healthy controls; also, MDMs from five healthy controls were in vitro HIV-1-infected in the absence or presence of 25HC. IFN-producing plasmacytoid dendritic cells (pDCs) were augmented in HESN than healthy controls in unstimulated and in vitro HIV-1-infected PBMCs. The expression of CH25H and several genes involved in cholesterol metabolism (ABCA1, ABCG1, cytochrome P450 family 7 subfamily b member 1, CYP7B1, LXRα, oxysterol binding protein, OSB.P., peroxisome proliferator-activated receptor-gamma, PPARγ, and SCARB1) were increased. These results were correlated with reduced susceptibility to in vitro HIV-1-infection of PBMCs and MDMs. Remarkably, the 25HC added to MDMs caused an increased cholesterol efflux and augmented resistance to in vitro HIV-1-infection [106].

Recently, there has been an augmented interest in finding polymorphisms in genes involved in lipid metabolism, which may affect susceptibility to developing dyslipidemia, lipodystrophy, and MetS complications in untreated and HIV patients on HAART to predict the clinical progression of the disease. A study conducted in Spain showed that single nucleotide polymorphisms (SNPs) in genes associated with atherogenic dyslipidemia, particularly the variants rs3135506 and rs662799 of the apolipoprotein-A5 (APOA5) gene, can influence the CD4 T-cell levels in chronic HIV-infected patients [107]. Similarly, a case-control study in Ghana found single nucleotide polymorphisms in four candidate genes and their association with dyslipidemia among ART-treated HIV patients, main variants in APOA5 (rs662799) and LDLR (rs6511720), respectively [108]. In research with HIV patients with HAART from Brazil, the SNPs (rs2273773 T > C, rs12413112 G > A, rs7895833 A > G, rs12049646 T > C) from the sirtuin-1 (SIRT1) gene, a modulator of transcription factors involved in energy regulation, were not associated with lipodystrophy and MetS, but white individuals with prolonged HAART intake were more prone to develop lipodystrophy [109]. A systematic review showed no association between the polymorphisms in APOC3 and PPAR-γ with lipodystrophy; although, there was an association between the G allele of the homeostatic iron regulator (HFE) and protection against the development of lipoatrophy when compared with the reference C allele. Conversely, the T allele of the estrogen receptor-2 (ESR2) gene was linked with the development of lipoatrophy when contrasted with the reference C allele. Besides, the GG genotype and the G allele of matrix metalloproteinase-1 (MMP1) were correlated with lipodystrophy in HIV patients receiving HAART [110].

Liver lipid accumulation induced by HIV infection deserves special attention, besides the glucose and lipid metabolism derangements which mean the risk of cardiovascular disease. Liver illness is the second most common cause of death in HIV-infected people; therefore, the role of HIV matrix protein p17, a multifactorial protein that mediates viral replication and the infection of immune T-cells and monocytes, besides the hepatic stellate cells, was described [111]. Protein p17 enhanced the expression and transcriptional activity of LXR and its coactivator, the mediator complex subunit-1 (MED1), via the activation of Jak/STAT signaling, which resulted in hepatic lipid accumulation via activation of the LXR/SREBP1c (sterol regulatory element-binding transcription factor 1) lipogenic pathway. In addition to non-alcoholic fatty liver disease and steatohepatitis in HIV patients, another hepatopathy has been overlooked, the gallstone disease. A recent African study evaluated the hepatic expression of LDLR, HMGCR, ABCA1, and transcriptional regulators of these genes (microRNA-148a, SREBP2) in HIV-positive patients on ART presenting gallstones. The HIV-positive group had higher total cholesterol with significantly elevated LDL-C levels than uninfected controls, while the scavenging LDLR for LDL-C was significantly decreased in the HIV group. The transcriptional regulator of LDLR, SREBP2, was decreased in HIV-positive patients; besides this, the regulatory miR-148a-3p was reduced with a concomitant increase in target ABCA1, regulating cholesterol efflux [112].

Metabolic comorbidities of both HIV infection alone and ART-induced ones include inflammation, dyslipidemia, atherosclerosis, MetS, lipodystrophy, myocardial disorders, diabetes, impaired hematopoiesis, cognitive damage, and liver injury. As previously commented, most of these depend on the cholesterol metabolism, which is altered by the HIV infection process itself and can be enhanced by HAART. Currently, there is a growing body of evidence pointing out the crucial role of Nef-induced impairment of genes involved in the cholesterol metabolism and the cholesterol-enriched regions of the plasma membrane, known as lipid rafts, which facilitate the HIV replication and the successful entry/exit of HIV in the target cells [113]. Nevertheless, Nef is not the only protein involved in the metabolic disorders observed in HIV patients; the HIV protein Vpr inhibits the PPARγ leading to lipotoxicity in mouse models [114] or, as mentioned above, the protein p17 that mediates liver steatosis. Another comorbidity is alcohol use in HIV patients. Because the in vitro and in vivo oxidative stress provoked by alcohol reduced the expression of antioxidant enzymes such as glutathione synthetase (GSS), superoxide dismutase (SOD), and glutathione peroxidase (GPx), as well as lowered ABC-transporters and the SREBP-2 transcription; while increased membrane lipid rafts caveolin-1 (Cav-1), and up-regulated HMGCR and cyclooxygenase-2 and 5-lipoxygenase (5-LOX), resulting in high levels of pro-inflammatory prostaglandin-E2. In sum, oxidative stress exacerbates the injury caused by HIV infection in alcohol consumers (Figure 2) [115].

### 2.4. Increased Oxidative Stress in People HIV+

OS is understood as the imbalance between the production of ROS/reactive nitrogen species (RNS) and other prooxidant chemical species and the antioxidant system in the body. ROS and RNS affect the cytochrome p450 enzyme system [116], required for chemical defense or detoxification, and the cellular response system of molecular signals [117]. Awareness of chemical prooxidant species and antioxidants has been gaining interest in recent decades because of their significant role in different signaling pathways, cell homeostasis maintenance, phagocytosis processes, immunological functions, vascular processes, and cell membrane stability [118,119]. ROS are generated naturally by aerobic cellular metabolism when oxygen is partially reduced, producing hydrogen peroxide (H_2_O_2_), superoxide anion (O_2_•^–^), free oxygens (^1/2^O_2_), and hydroxyl radicals (OH•). Species from nitrogen, iron, copper, and sulfur substances are equally found [120,121]. When the levels of prooxidant species are not removed or processed by the endogenous antioxidant system, oxidative processes are activated and damage biomolecules such as proteins, carbohydrates, lipids, and DNA, being factors for the development of cardiovascular diseases, diverse types of cancer, neurological conditions, and other diseases featured by the presence of inflammatory processes [122,123].

Other frequent forms of ROS production are through the electron transport chain, degradation of lipids and amino acids, and protein folding in the lumen of the endoplasmic reticulum (ER). ROS production is mediated by the nicotinamide adenine dinucleotide phosphate (NADPH) oxidase (NOX) family complex as NOX1, NOX2, NOX4, and NOX5 [5,124]. NOX members are found in muscle tissue and represent the primary sources of ROS; its activity has been investigated in conditions such as hypertension, hypercholesterolemia, and endothelial dysfunction [110,111,112,125,126,127]. Another enzyme that generates ROS is xanthine oxidoreductase, principally found in the liver and gut, although distributed in smaller proportions in the lungs, kidneys, heart, and brain plasma. Among its functions are redox reactions of sulfhydryl groups through intrasubunit disulfide bridges that allow the conversion of NAD^+^-dependent xanthine dehydrogenase (XDH) into oxygen-dependent xanthine oxidase (XO). Furthermore, both XO and XDH can oxidize NADH with the concomitant formation of ROS [5,124].

In vitro and in vivo studies and HIV clinical trials have highlighted the presence of OS, mainly by the formation of ROS in monocytes [128,129,130]. Due to the virus replication processes, the organism must preserve its capacity to clear ROS to avoid CD4 lymphocytes depletion, an subsequent elevation in viral load, and the oxidation of guanine, which results in the formation of 8-oxoguanine (8-oxoG), one of the most important markers of DNA oxidation and which is related to the development of cancer and neurological diseases (Alzheimer’s and Parkinson’s) [131]. OS is also incremented in people with HIV by the viral Tat protein (trans-activator), which promotes the generation of a superoxide anion in the mitochondria, triggering DNA damage and activating oncogenes [132]. Concomitantly, a depletion in total antioxidant capacity in plasma is associated with decreased levels of CD4 and CD8 lymphocytes and the endogenous antioxidant glutathione (GSH) [133]. However, the redox imbalance is higher in patients without ART due to the action of the drugs to preserve or increase the number of CD4 lymphocytes, considering that ART also exacerbates the oxidant state in the organism [124]. Therefore, the risk of lymphocyte reduction and the maintenance of OS in HIV patients is latent, becoming a factor for developing other diseases or complicating some symptoms in the typical course of the disease.

Given these conditions, natural therapies based on foods with high antioxidant capacity seem to represent an efficient option to prevent and diminish the organic functional deterioration caused by the OS excess generated by HIV and ART.

### 2.5. Antioxidants and Phytochemicals

Antioxidants are substances present at low concentrations compared to those of an oxidizable substrate, which remove, retard, or prevent the oxidation of the substrate. Antioxidants, when interacting with a free radical, yield an electron to it, oxidizing it in turn and transforming it into a relatively stable, non-toxic molecule which, in some cases, can be regenerated to its reduced form by the action of other antioxidant systems (for example vitamin E) [134,135].

The nature and structure of antioxidants are diverse, and they are traditionally classified into endogenous and exogenous antioxidants. There are those belonging to the superoxide dismutase (SOD) family among the endogenous antioxidants, such as manganese SOD (MnSOD), the only enzyme of this family found inside the mitochondria [136], copper-zinc SOD (CuZnSOD), which acts in cytoplasm reducing superoxide anion [137], and extracellular SOD (ecSOD), the principal regulator of nitric oxide (NO) in the vasculature [138]. Other essential antioxidant enzymes are catalase (CAT), which promotes the breakdown of H_2_O_2_ and sulfur oxidase or sulfur reductase activity [139], selenium-dependent glutathione peroxidase (SeGPx); also, the thioredoxin system, an NADPH-dependent enzyme, thioredoxin reductase (TrxR) implicated in DNA and protein repair [140]. In addition, TrxR serves as a scavenger of transition metals (iron, copper, and silver) and non-enzymatic metabolites with antioxidant capacity (GSH, urate, bilirubin, ubiquinones). Exogenous antioxidants enter through the food chain and require continuous renewal. Natural exogenous antioxidants include vitamins (E and C), lipoic acid, selenium, phytochemicals β-carotenes, vitamin A, ellagic acid, and flavonoids [141].

Among synthetic antioxidants are the transition metal chelators (deferoxamine, α-keto-hydroxy-pyridines), ROS scavengers as 21-amino-steroids, 2-methyl-aminochromanes, pyrrolopyrimidines, butylated hydroxytoluene, phenyl-tert-butyl-nitrone, n-acetyl-cysteine, nonsteroidal anti-inflammatory drug, probucol, β-blockers, calcium-channel blockers, angiotensin-converting enzyme inhibitors, and xanthine oxidase inhibitors (allopurinol), antioxidant enzymes for therapeutic use (pyran-SOD, desferal/MnIII, Fe-TPEN, Fe-TPAA, EUK-8, M40403, ebselen), NADPH oxidase inhibitors (oxatomide), and trace elements (zinc, iron, copper, selenium, magnesium) [142]. 

#### 2.5.1. Vitamin A

Vitamin A or retinol is ingested as retinyl esters or carotenoids and metabolized to 11-cis-retinal, all-trans-retinoic acid, which is the principal mediator of the biological actions of the vitamin A and binds to retinoic acid receptors (RAR), which heterodimerize with retinoid X receptors (RXR). RAR-RXR heterodimers function as transcription factors, binding RAR responsive elements in promoters of different genes [143,144]. The role of vitamin A in various chronic non degenerative diseases has been highlighted, principally for its activity on inflammatory pathways and effects on oxidative stress [145,146]. It is recognized that the storage and uptake of vitamin A are principally regulated by hepatocytes (10–20% of the stored vitamin) and non-parenchymal liver stellate cells (80–90% of the stored vitamin), which are responsible for the mobilization of the vitamin A from the liver. Therefore, some animal and human studies have focused on assessing the importance of vitamin A intake in liver diseases during their initial stages (e.g., non-alcoholic fatty liver disease) [147,148].

#### 2.5.2. Carotenoids

Carotenoids are tetraterpenes consisting of multiple isoprenoid units with an unsaturated cyclohexane ring at each end; these phytochemicals are known as natural pigments that provide yellow, orange, and red colorations to foods. These lipophilic molecules possess the property of absorbing light [149]. The polyene chain of carotenoids is highly reactive and rich in electrons that quickly form short-lived free radicals in the presence of oxidizing substances. Some of the most critical carotenoids whose source is food include α-carotene, β-carotene, lycopene, curcumin, and lutein, which may serve as biomarkers to determine vegetables and fruit intake and clarify the diet–disease relationship [150]. About six hundred types of carotenoids have been identified in plants, fungi, algae, and bacteria; humans and animals cannot synthesize them, so their principal source comes from diet, and about 20 carotenoids have been found in blood plasma and tissues, and some of them, such as lycopene, owing to their lipophilic nature, are transported by lipoproteins [151]. Carotenoids prevent oxidative damage to various molecules, specifically lipoproteins and other lipolytic compartments [152]. The mechanism of action of carotenoids in human health has not been clarified. A hypothesis is that it serves as a potent antioxidant suppressing the superoxide radical [153,154] and decreasing oxidative stress. Like some other antioxidants, it can maintain vascular homeostasis by regulating TNF-α-mediated inflammation [155], reducing the processes of apoptosis, autophagy, and cell necrosis in neurodegenerative diseases [156]. Its properties in health also include decreasing the growth of malignant tumors and modulating gene expression and immune system response [157].

#### 2.5.3. Flavonoids

Flavonoids are natural polyphenolic metabolites of high abundance in plants, fruits, and vegetables, providing health benefits because of their high antioxidant, anti-inflammatory, antiviral, and anticarcinogenic activities [158]. All flavonoids possess a similar structure consisting of two aromatic rings bonded to three carbon atoms [159]. Flavonoids are typically classified into anthocyanidins, isoflavones, catechins, flavonols, and flavanones and are reported to act as cardioprotective agents by inhibiting LDL oxidation enhancing vasodilation, and improving the lipid profile [160]. Besides its ability to modulate the immune system and decrease the viral load in diseases like HIV and SARS-CoV-2, the mechanisms proposed can bind to the virus wall or slow the replication process, but this is still a field of research [161,162,163].

#### 2.5.4. Ellagic Acid

Ellagic acid is a component of hydrolyzable tannins known as ellagitannins. When metabolized by the intestinal microbiota, this phytochemical forms urolithins A, B, C, or D, whose study is interesting as modulators of OS and its anti-inflammatory properties, mostly [164]. The sources are nuts and some seeds in food, but it is mainly found in fruits at high concentrations, including blackberries, raspberries, cherries, and the red pomegranate [165,166]. In recent years, ellagic acid’s role in MetS, diabetes, obesity, and cardiovascular conditions, including hypertension, atherosclerosis, and endothelial dysfunction, has been investigated, alongside its properties in reducing OS, enhancing lipid profile and cholesterol metabolism, decreasing inflammatory processes, and improving insulin secretion and glucose uptake [167,168]. Many authors point out that these properties are attributable to its ability to decrease the plasmatic concentration of triglycerides and LDL-C but increase HDL-C, enhance NO’s bioavailability, and raise the expression of hepatic LDL receptors [169,170,171,172].

### 2.6. Role of Antioxidants on Lipid Metabolism during HIV Infection

The pathogenesis of dyslipidemia in HIV patients remains poorly understood. Molecular mechanisms are focused on the increased inflammation pathways, and it is known that HAART led to lipid abnormalities such as dyslipidemia. By investigating genes involved in lipid and glucose metabolism alterations in HIV-infected patients treated with HAART, the transcriptional profiling showed an overexpression of MnSOD and heat shock protein (HSP27) in patients with MetS and HAART [173]. That suggests the activation of an endogenous response to the deleterious effects of HAART (Figure 3).

The chronic use of cART with two nucleoside analog inhibitors-NRTIs increases OS and hyperlipidemia caused by lipodystrophy [174]. In HIV-1 transgenic rats, the accumulation of total cholesterol in the liver and serum hypertriglyceridemia is concomitant with higher expression of SREBP-1 in animals treated with cART. However, the enhanced expression of SREBP-1 was suppressed by Mg-supplementation, which was associated with the reduction in serum cholesterol and triglyceride levels [175].

Antioxidants are well known to modulate transcription factors to exert antioxidant and anti-inflammatory effects. The NF-κB/TNF-α pathway was the first inflammatory pathway identified in people with AIDS [176,177]. The use of chain-breaking, lipid-soluble, phenolic antioxidants, such as butylated hydroxyanisole (BHA), ordihydroguaiaretic acid (NDGA), or alpha-tocopherol (vitamin E), can inhibit NF-κB activation because they are peroxyl radical scavengers. BHA also suppresses HIV-enhancing activity through inhibition of NF-κB in lymphoblastoid T-cell and monocytic lymphoblastoid cultures. However, this effect is only achieved if the cells are in an appropriate redox state [176], so the use of compounds that have this duality (antioxidant and anti-inflammatory response) could be a promising therapy. 

TNF-cachectin enhances HIV expression and correlates positively with serum T.A.G. levels [177,178], suggesting an excellent pharmacological target for HIV dyslipidemia. In 1993, Dezube et al. showed that pentoxifylline (PTX) reduces TNF-α activity and HIV replication in cultured cells and HIV patients (n = 17). The 8-week study in HIV patients with consumption of PTX (400 mg) reduced TNF-α mRNA and serum T.A.G. levels [177]. PTX is a methylxanthine compound with anti-inflammatory activity inhibiting the NF-κB/TNF-α pathway. It has an antioxidant response by activating the nuclear factor erythroid 2-related factor 2 (NRF2), a transcription factor that regulates the gene expression of endogenous antioxidant enzymes linked to GSH metabolism [179,180].

NRF2 plays a vital role in adipogenesis. Its deficiency in the adipose tissue of ob/ob mice leads to MetS with the aggravation of insulin resistance, hyperglycemia, and hypertriglyceridemia caused by Tat protein, which is one of the six regulatory proteins required for HIV-viral replication [181]. Subsequently, it was described that Tat protein decreased intracellular GSH levels and increased ROS production by enhancing NRF2 expression in MAGI cells. However, using N-acetylcysteine or the overexpression of NRF2 suppressed Tat-induced HIV-1 LTR transactivation, leading to its replicative activity against HIV-1 [182]. This effect was found with the use of tanshinone II-A, a lipid-soluble major monomeric derivative of *Salvia miltiorrhiza* (Danshen) root that increases GSH levels by regulating NRF2 activation and SIRT1 activity in TZM-bl cells (Figure 3) [183]. Therefore, adequate regulation of NRF2 activity is necessary to maintain intracellular redox homeostasis during harmful signals by HIV infection.

The regulation of NRF2 has been widely studied in several models of metabolic diseases. The use of oleanolic acid in diabetic Lep db/db mice activates Nrf2, increasing the glutamate-cysteine ligase gene and GSH synthesis, with a concomitant reduction in serum triglycerides, total cholesterol, LDL-C, HDL-C, and free fatty acids, as well as the increased levels of HDL-C in serum, and a reduction in hepatic lipid accumulation [184,185]. Additionally, vitamin E supplementation (50 mg/kg) in a model of atherosclerosis in male albino rabbits with high cholesterol diet (2%) showed a decrease in matrix metalloproteinase-1 expression and an increase in PPARγ and levels of the glutathione S-transferases GSTα and ABCA1 in rabbit aorta through the induction of Nrf2-mediated antioxidant genes, which was associated with a decrease in lesion progression (Figure 3) [186]. In this regard, antioxidant activators of Nrf2 such as pterostilbene (PTS) have been evaluated in streptozotocin (STZ)-diabetic animal models showing regulation of VLDL-, LDL-C, and HDL-C levels in serum and reducing lipid peroxidation with the induction of SOD, CAT, GPX, and GSH synthesis through Nrf2 activation in the liver [187]. Therefore, these data suggest that NRF2 induction by some exogenous antioxidants could be a potential strategy used to decrease concomitant effects in HIV patients. Further clinical trials should be conducted to evaluate this hypothesis.

One such novel approach to improving HDL-C function involves agents that increase the activity of lecithin cholesterol acyltransferase (LCAT), an enzyme involved in HDL-C maturation and reverse cholesterol transport. In 118 HIV-infected patients aged 19–71 years who had received HAART for 6–24 months, six SNPs in the LCAT gene were associated with dyslipidemic outcomes [188]. Although the use of some antioxidants which can modulate LCAT expression in HIV-infected patients has not been proven, in a clinical trial in healthy subjects, the intake of functional virgin olive oil enriched its phenolic compound, increased LCAT and paraoxonase enzymatic activity, and this was associated with increased levels of HDL-C [189].

The use of some antioxidants could downregulate the NLRP3 inflammasome. For example, in streptozotocin-diabetic rats, the use of quercetin and allopurinol for 7 weeks inhibits the activation of NLRP3, PPARa, and up-regulation of sterol regulatory element-binding protein-1c (SREBP-1c) by TXNIP down-regulation in the liver, showing less hepatic lipid accumulation [190]. Quercetin also reduced insulin resistance and hyperlipidemia through the downregulation of AMPK/TXNIP and subsequent inhibition of NF-κB pathway/NLRP3 inflammasome activation in the hypothalamus of rats fed with 10% fructose and quercetin (50 and 100 mg/kg) [191]. 

The benefits of quercetin have been proved also in the cancer context by its potential ability to regulate the cell cycle, tyrosine kinase inhibition, and AMPK pathways, which are important for the regulation of inflammatory processes. However the major limitation of the use of polyphenols is their low bioavailability after ingestion, therefore some alternatives are rising to improve their biological activity to reach target cells. For example, Quagliariello et al. (2015), developed a hyaluronic acid hydrogel loaded with quercetin to inhibit overexpression of Aurora kinases to reduce cell proliferation in human thyroid cancer [192]. In this sense, a recent investigation showed that epigenetic changes in aurora kinase B (AURKB) and aurora kinase C (AURKC) are involved during the early stages of HIV-1 infection [193]. Therefore, it will be interesting to develop clinical trials with hyaluronic acid and quercetin hydrogel.

Other alternatives to improve intestinal bioabsorption include the encapsulation of bioactive compounds in liposomes to enhance plasma concentration and delivery in plasma. For example, in macrophages derived from healthy and HIV-infected individuals, the treatment with a liposomal formulation of GSH reduced free GSH and correlated with less intracellular growth of *Mycobacterium tuberculosis* and free radicals and the immunosuppressive cytokines IL-10 and TGF-β [194]. The use of liposomes is an emerging field of research and it could be good tool to be investigated to reduce inflammatory processes. Quagliariello et al. (2015), found that the use of liposomes with butyric acid and chitosan increases their ability to be internalized in hepatoblastoma cells (HepG2) and to inhibit IL-8, IL-6, TNF-α, and TGF-β expression [195], and their use in treating dyslipidemia in HIV infection context; this is a novel field of research. It is a new field of research to evaluate antioxidant compounds that can activate endogenous antioxidant batteries or directly increase the activity of antioxidant, anti-inflammatory, and antilipidemic enzymes.

## 3. Conclusions

HIV infection and ART contribute to dyslipidemia and lipodystrophy in HIV patients. The evidence shows strong evidence of the participation of OS and inflammatory component in cardiometabolic worsened disease, and the use of natural compounds with anti-inflammatory and antioxidant action could be a promising therapy. The activation of gene transcriptional factors by antioxidants to regulate cholesterol efflux and fatty acid distribution in the body is still a field of research. However, due to the molecular complexity of the HIV infection, future study designs should include preclinical evidence of target genes that regulate cholesterol efflux without interfering with lymphocyte count and evaluating the genetic and dietary background of patients. Pharmacogenomic and nutrigenomic studies and other omics tools are necessary to evaluate the effect of the use of dietary antioxidants and the genomic background among populations.

## Figures and Tables

**Figure 1 ijms-23-05592-f001:**
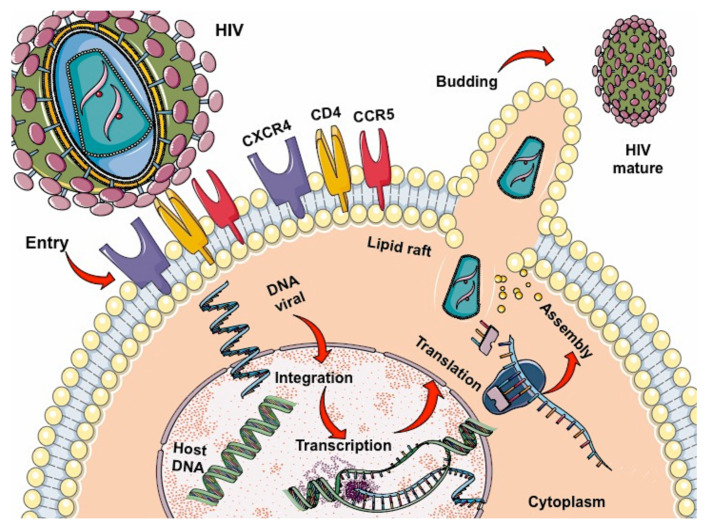
Lifecycle of HIV: binding, fusion, reverse transcription, integration, replication, assembly, budding, and maturation.

**Figure 2 ijms-23-05592-f002:**
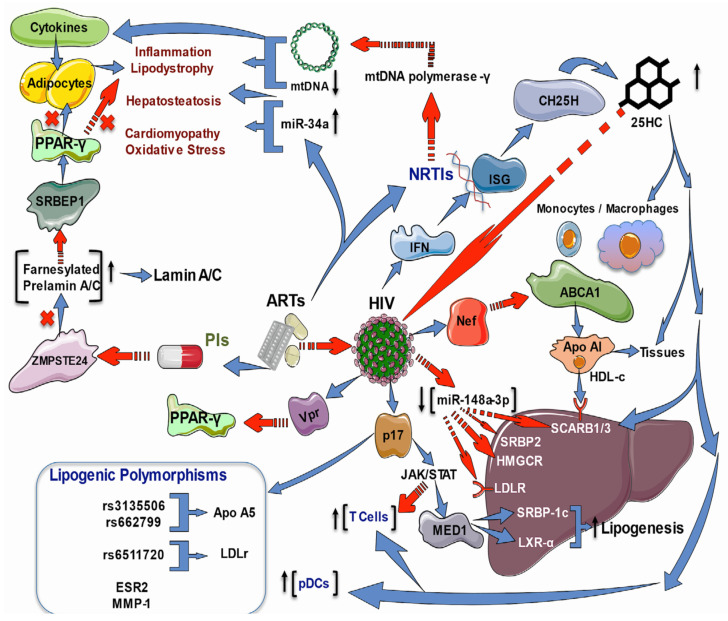
Effects of ART and HIV infection on lipid and cholesterol regulatory genes. ART may cause lipodystrophy in HIV patients administered with PIs or NRTIs. PIs inhibit ZMPSTE24, which processes the farnesylated prelamin-A/C, inducing its accumulation. Prelamin-A/C sequesters S.R.E.B.P., decreasing its activity on PPAR-γ, impairing the regulation of adipogenesis transcription, and promoting the HIV-associated cardiomyopathy by NF-kB-induced inflammation. NRTIs inhibit mtDNA polymerase-γ transcription, leading to mtDNA depletion, which causes lipodystrophy and hepatosteatosis mediated by pro-inflammatory cytokines. Besides this, ART increases miR34a, promoting hepatosteatosis, cardiomyopathy, and OS HDL-C with Apo1 participates in the RCT from peripheral tissues into circulation and the liver. HIV infection via Nef harms the monocyte-macrophage cholesterol efflux by increasing ABCA1 degradation; also, the downregulation of genes related to cholesterol uptake (LDLR and SCARB1/3), synthesis (HMGCR), and regulation (SREBP2 and LXRα). The HIV matrix protein p17 enhances the expression and transcriptional activity of LXR, and its coactivator (MED1), via the activation of Jak/STAT signaling, which results in hepatic lipid accumulation via activation of the LXR/SREBP1c lipogenic pathway and mediates liver steatosis. HIV-positive patients on ART present gallstones and higher total cholesterol with significantly elevated LDL-C levels but decreased scavenging LDLR for LDL-C. The transcriptional regulator of LDLR, SREBP2, is decreased in HIV infection; besides this, the regulatory miR-148a-3p is reduced with a concomitant increase in target ABCA1. Additionally, the HIV protein Vpr inhibits the PPARγ leading to lipotoxicity. Antiviral IFNs upregulate ISGs (CH25H), which converts cholesterol to 25HC that inhibits the growth of enveloped HIV by blocking membrane fusion with cells, suppresses the HIV replication, and increases the number of T cell and pDCs; it also augments the expression of genes involved in cholesterol metabolism (ABCA1, ABCG1, CYP7B1, LXRa, OSB.P., PPAR-γ, and SCARB1/3). The SNPs rs3135506 and rs662799 of the APOA5 gene and rs6511720 of the LDLR gene were associated with the development of atherogenic dyslipidemia. The T allele of ESR2 and G.G. genotype of MMP1 were found to be associated with lipoatrophy.

**Figure 3 ijms-23-05592-f003:**
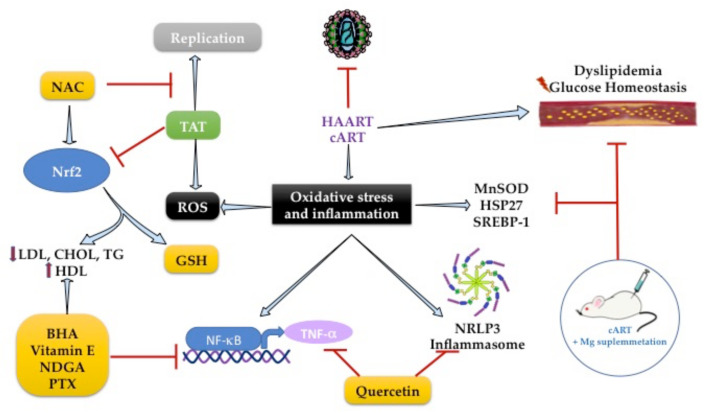
Gene targets antioxidants with antilipidemic effects to reduce oxidative stress and inflammation in HIV infection.

**Table 1 ijms-23-05592-t001:** Protease Inhibitors used as antiretroviral in children and adults.

ART	Pediatric Dose	Adult Dose	Advantages	Disadvantages	Ref.
Saquinavir (SQV, Invirase)	350 mg/12 h	600 mg/8 h	Choice as antiretroviral in pregnancy and minimally secreted in breast milk. Efficient hepatic secretion (88%).	Gastrointestinal intolerance, headache, increased transaminases. The safety and activity of saquinavir combined with ritonavir in pediatric patients under two years of age are not established. Risk of arrhythmias and hypertension.Chronic consumption increased plasma cholesterol and triglyceride levels	[37,38,39]
Ritonavir (Norvir)	>two years 400 mg/100 mg/12 h	600 mg/12 h	Better absorption in lymphoid tissue can be taken together with food and generates an improved treatment tolerance.	Long-term gastrointestinal problems, pancreatitis, paresthesias, increased transaminases, asthenia, hepatitis, and palate alteration. Alteration of genes expression related lipid metabolism (CYP7A1, CITED2 and G6PC)	[40,41,42]
Indinavir (Crixivan)	>three years 500 mg/m^2^/8 h	800 mg + ritonavir 100 mg/12 h.	Bioavailability of 60%. Used in association with other antiretrovirals to delay disease progression and reduce the risk of opportunistic infections.	Decreases gastric pH, short half-life (3 administrations per day), presenting dietary restrictions (fasting or bland food). Development of nephrolithiasis, so abundant liquid consumption is essential. Increased plasma cholesterol, glucose, and triglyceride levels	[43,44]
Nelfinavir (Viracept)	45–55 mg/kg/12 h 25–35 mg/kg/8 h	750 mg/8 h	The antiviral effect is prolonged for at least 21 months. Bioavailability increases when combined with food.	Conditions including skin rash, allergic reactions, hepatitis, abnormalities in liver function tests, nausea, vomiting, diarrhea, abdominal pain, fatigue, fever, headache, and myalgia may appear. Long-term use can produce Stevens–Johnson syndrome and toxic epidermal necrolysis. It readily crosses the placental barrier, and its presence in breast milk has been reported. Related with apoptosis and necrosis by increasing ROS production	[45,46,47]
Amprenavir (Lexiva)	20 mg/kg dos veces al día o 15 mg/kg 3 veces al día	1200 mg/12 h	Improved dosing schedule for twice-daily administration with no restrictions on meal times or fluids. Absorption is increased after oral administration. The bioavailability of the solution is 86% compared to caplet formulation.	Owing to its formulation, vitamin E supplementation is avoided. It is not recommended for people with renal or hepatic insufficiency. Changes in the lipid profile by developing hypertriglyceridemia or hypercholesterolemia.	[48,49]

**Table 2 ijms-23-05592-t002:** Nucleoside and nucleotide reverse transcriptase inhibitors (NRTIs).

ART	Pediatric Dose	Adult Dose	Advantages	Disadvantages	Ref.
Zidovudine (Retrovir)	Infants 4–9 kg: 12 mg/kg/12 h Infants 9–30 kg: 9 mg/kg/12 h	300 mg/12 h	Combined with IFN-α prevents toxic side effects. Safety during pregnancy For use as first-line prophylaxis of infection in newborn infants.	Produces lactic acidosis usually associated with hepatomegaly and hepatic steatosis. Treatment with zidovudine is associated with the appearance of lipoatrophy. Long-term consumption can lead to osteonecrosis, anemia, and neutropenia.Elevated oxidative and endoplasmic reticulum stress resulting in lipid acummulation	[52,53,54,55]
Didanosine (Videx)	>90 days age: 120 mg/m^2^/12 h >6 years 240 mg/m^2^	>60 kg 200 mg/12 h <60 kg 125 mg/12 h	Replacement for people intolerant to zidovudine	Combined with stavudine leads to lactic acidosis, pancreatitis, lipoatrophy, and hepatic dysfunction.	[56,57]
Zalcitabine (Hivid)	>4 years 500 mg/m^2^/8 h	Combinated 800 mg + 100 mg ritonavir/12 h	Stable at gastric pH and shows reliable bioavailability (approximately 70% to 90%). It is considered ten times more potent than zidovudine (AZT) on a molar basis in vitro.	Associated with the development of peripheral neuropathy, the incidence of anemia, leukopenia, neutropenia, and elevated glutamic-oxaloacetic and glutamic-pyruvic transaminases.	[58,59]
Stavudine (Zerit)	<30 kg 1 mg/kg twice daily	<60 kg 30 mg twice daily >60 kg 40 mg twice daily	Powder presentation for pediatric patients under three months of age and adults with swallowing problems and dysphagia.	Medium- and long-term administration produces lactic acidosis, lipoatrophy, and polyneuropathy. Increased total cholesterol, LDL-C, and triglycerides	[60,61,62]
Lamivudine (Epivir)	150 mg/ 12 h	300 mg/24 h	The oral suspension enhances the drug administration for children over three months of age and weighing less than 14 kg or for patients with dysphagia. Absolute bioavailability is close to 82% and 68% in adults and children. Potent antiviral activity against chronic hepatitis B and HIV	It is not recommended as monotherapy.Administration of this antiretroviral can lead to pancreatitis, hepatitis, anemia, thrombocytopenia, neutropenia, and alopecia. Combined short-term decreased HDL-C and increased total cholesterol	[63,64]
Abacavir (Ziagen)	≥3 months 8 mg/kg/12 h	300 mg/12 h 600 mg/24 h	Dosage is once a day. Very low toxicity	It is contraindicated in patients with end-stage renal disease and not recommended in pregnant women. Produces lactic acidosis. Increased total cholesterol at short-term.	[65,66,67]

**Table 3 ijms-23-05592-t003:** Non-nucleoside reverse transcriptase inhibitors (NNRTIs).

ART	Pediatric Dose	Adult Dose	Advantages	Disadvantages	Ref.
Delavirdine (Rescriptor)	10–47 kg 15 mg/kg/day	800 mg/24 h	Administer without food restriction. No interaction with proton pump inhibitors. No inhibition and non-induction of CYP450 and CYP34. One-time daily administration is allowed.	Consumption of this product could result in headaches, hypophosphatemia, hypomagnesemia, hypertension, dyspnea, and aminotransferase elevation. It is not suitable for use during pregnancy. Increased plasma cholesterol levels by CYP27A1 inhibition	[69,70]
Efavirenz (Sustiva)	>10 kg 200 mg/24 h	>40 kg 600 mg/24 h	Dosage is once a day	It is not administered under three months of age and as monotherapy mainly for its adverse reactions involving the nervous system. Increased cholesterol through activation of PXR and overexpression of lipogenic genes.	[71,72,73]
Etravirine (Intelence)	16–20 kg 100 mg/12 h. >25 kg 125 mg/12 h	200 mg/12 h	Dissolves in water for easy administration. Suitable for use during pregnancy.	Long-term consumption of this medication causes osteonecrosis, rash, diarrhea, and nausea. Changes in redox system modifying catalase, glutathione peroxidase, and superoxide dismutase activity.	[74,75,76]
Nevirapine (Viramune)	>8 years 120 mg/12 h	200 mg/12 h	Bioavailability of 90%. Efficient in the prevention of mother-to-child HIV transmission.	Long-term administration can cause skin rash and hepatic toxicity. Not used during lactation. Combined with another ART decreased HDL-C and increased LDL-C, total cholesterol, and triglycerides	[77,78,79]
Rilpivirine (Edurant)	<8 years 200 mg/m^2^24 h	400 mg/24 h	Suitable for use during pregnancy	Long-term administration produces severe skin rash and lipodystrophy. Less effective in lipid profile regulation compared to other NRTIs	[64,80,81]

## Data Availability

Not applicable.

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
