# Peer review of "Antiretroviral Therapy-Induced Dysregulation of Gene Expression and Lipid Metabolism in HIV+ Patients: Beneficial Role of Antioxidant Phytochemicals"

_ijms, 2022, doi:10.3390/ijms23105592_

Round 1

Reviewer 1 Report

This review described the relationship between prolongation of ART and oxidative stress (OS) and alterations in lipid metabolism, and the natural products that can help diminish this effect.

Questions and comments:

  1. Whether the tables can show more content about the relationship of anti-HIV drugs and OS/alterations in lipid metabolism?
  2. It looks like too many things about the basic HIV, AIDS, and ART (page 1-8, which is already well documented), can this part be shortened?

Minor change:

Line 95 should change to “chemokine co-receptors CCR5 or CXCR4 co-receptors found mainly on”

Author Response

Response to Reviewer 1 Comments

Point 1: Whether the tables can show more content about the relationship of anti-HIV drugs and OS/alterations in lipid metabolism?

Response: Information related to the modification of the lipid profile or the presence of oxidative stress in most of the ARTs mentioned in the tables was added, with new supporting literature.

Point 2: It looks like too many things about the basic HIV, AIDS, and ART (page 1-8, which is already well documented), can this part be shortened?

Response: Some information was shortened in terms of the life cycle of HIV, AIDS and ART, keeping information specific but also offering general knowledge useful for non-expert readers who are new to this field.

Minor change:

Line 95 should change to “chemokine co-receptors CCR5 or CXCR4 co-receptors found mainly on”

Response: Word co-receptors was repositioned

*Comment: The three figures presented are originals, neither any of its elements were copied from other pictures published on the Internet or from scientific articles.

Reviewer 2 Report

Review titled "Antiretroviral therapy-induced Dysregulation of Gene Expression and Lipid Metabolism in HIV+ Patients: Beneficial Role of Antioxidant Phytochemicals" described the potential benefits of nutraceuticals in HIV patients. The overall structure of the manuscript is of good quality, introduction, methods and conclusions are well performed and references are of good quality and updated on this field. However, authors should improve the manuscript in several parts:

1) Authors should improve the description of the main pathways involved in nutraceutical benefits in these patients, what about the NLRP3 inflammasome and myddosome?

2) Authors should describe the potential hepatoprotective and cardioprotective properties of SCFA like butirrate also in cancer patients and HIV patients ( cite 10.3892/or.2018.6932 )

3) Authors should at least introduce the potential of reishi, cordyceps and grifola as Complementary and alternative medicines in patients with HIV or cancer ( cite 10.18632/oncotarget.24984 )

4) Authors should explain how quercetin could be a beneficial nutraceutical and how it could reduce interleukins involved in HIV progression and patient survival. Moreover, authors should also decribe its main limitation after oral administration such as the low bioavailability and the updated strategies aimet to improve its anticancer, anti-inflammatory and antiviral properties for example through hyaluronic acid-based formulations ( cite 10.1002/jcp.25283).

Author Response

Response to Reviewer 2 Comments

Suggested corrections are shown in green in the document.

Point 1: Authors should improve the description of the main pathways involved in nutraceutical benefits in these patients, what about the NLRP3 inflammasome and myddosome?

Response: We included in section 2.3 (Effects of HIV infection and ART on lipid and cholesterol genes) the action of NLRP3 inflammasome in atherosclerosis development and in HIV infection. Also, the use of some compounds that down regulates NLRP3 inflammasome was discussed in section 2.6 (Role of antioxidants on lipid metabolism during HIV infection).

Point 2: Authors should describe the potential hepatoprotective and cardioprotective properties of SCFA like butirrate also in cancer patients and HIV patients ( cite 10.3892/or.2018.6932 )

Response: Thanks for the suggestion and the use of butyrate was included in section 2.6 as an alternative to improve intestinal biosorption of natural compounds during HIV infection and inflammation.

Point 3: Authors should at least introduce the potential of reishi, cordyceps and grifola as Complementary and alternative medicines in patients with HIV or cancer ( cite 10.18632/oncotarget.24984 )

Response: We appreciate the remark to include the information you kindly suggested as a new reference; nevertheless, we did not consider it given the length of this Review, which could be increased and possibly exceed the boundaries of readers' interest. We focus on the commented phytochemicals in the corrected version of the text, selected just for meeting the criteria of modulating gene expression related to cholesterol, oxidative stress, and lipid metabolism. Of course, we recognize that some other interesting secondary metabolites have possible effects on HIV infection and other pathologies wherein oxidative stress is implicated, regulating target metabolic and signaling pathways.

Point 4: Authors should explain how quercetin could be a beneficial nutraceutical and how it could reduce interleukins involved in HIV progression and patient survival. Moreover, authors should also describe its main limitation after oral administration such as the low bioavailability and the updated strategies aimet to improve its anticancer, anti-inflammatory and antiviral properties for example through hyaluronic acid-based formulations ( cite 10.1002/jcp.25283).

Response 2: The use of quercetin was included in section 2.6 as potential nutraceutical to targent NLRP3 inflammasome and cytokines. Also, it was very useful to introduce the discussion of the low bioavailability of many nutraceuticals. Therefore, the strategies included in section 2.6 were the use of liposomes and hydrogels to improve intestinal absorption.

*Comment: The three figures presented are originals, neither any of its elements were copied from other pictures published on the Internet or from scientific articles.